# Embed and Project:
# Discrete Sampling with Universal Hashing

**Stefano Ermon, Carla P. Gomes**
Dept. of Computer Science
Cornell University
Ithaca NY 14853, U.S.A.

**Ashish Sabharwal**
IBM Watson Research Ctr.
Yorktown Heights
NY 10598, U.S.A.

**Bart Selman**
Dept. of Computer Science
Cornell University
Ithaca NY 14853, U.S.A.

## Abstract

We consider the problem of sampling from a probability distribution defined over a high-dimensional discrete set, specified for instance by a graphical model. We propose a sampling algorithm, called PAWS, based on embedding the set into a higher-dimensional space which is then randomly projected using universal hash functions to a lower-dimensional subspace and explored using combinatorial search methods. Our scheme can leverage fast combinatorial optimization tools as a blackbox and, unlike MCMC methods, samples produced are guaranteed to be within an (arbitrarily small) constant factor of the true probability distribution. We demonstrate that by using state-of-the-art combinatorial search tools, PAWS can efficiently sample from Ising grids with strong interactions and from software verification instances, while MCMC and variational methods fail in both cases.

## 1 Introduction

Sampling techniques are one of the most widely used approaches to approximate probabilistic reasoning for high-dimensional probability distributions where exact inference is intractable. In fact, many statistics of interest can be estimated from sample averages based on a sufficiently large number of samples. Since this can be used to approximate #P-complete inference problems, sampling is also believed to be computationally hard in the worst case [1, 2].

Sampling from a succinctly specified combinatorial space is believed to much harder than *searching* the space. Intuitively, not only do we need to be able to find areas of interest (e.g., modes of the underlying distribution) but also to balance their relative importance. Typically, this is achieved using Markov Chain Monte Carlo (MCMC) methods. MCMC techniques are a specialized form of *local search* that only allows moves that maintain detailed balance, thus guaranteeing the right occupation probability once the chain has *mixed*. However, in the context of hard combinatorial spaces with complex internal structure, mixing times are often exponential. An alternative is to use complete or systematic search techniques such as Branch and Bound for integer programming, DPLL for SATisfiability testing, and constraint and answer-set programming (CP & ASP), which are preferred in many application areas, and have witnessed a tremendous success in the past few decades. It is therefore a natural question whether one can construct sampling techniques based on these more powerful *complete search methods* rather than local search.

Prior work in cryptography by Bellare et al. [3] showed that it is possible to uniformly sample witnesses of an NP language leveraging universal hash functions and using only a small number of queries to an NP-oracle. This is significant because samples can be used to approximate #P-complete (counting) problems [2], a complexity class believed to be much harder than NP. Practical algorithms based on these ideas were later developed [4–6] to near-*uniformly* sample solutions of propositional SATisfiability instances, using a SAT solver as an NP-oracle. However, unlike SAT,

most models used in Machine Learning, physics, and statistics are *weighted* (represented, e.g., as graphical models) and cannot be handled using these techniques.

We fill this gap by extending this approach, based on hashing-based projections and NP-oracle queries, to the weighted sampling case. Our algorithm, called PAWS, uses a form of approximation by quantization [7] and an embedding technique inspired by slice sampling [8], before applying projections. This parallels recent work [9] that extended similar ideas for unweighted counting to the weighted counting world, addressing the problem of discrete integration. Although in theory one could use that technique to produce samples by estimating ratios of discrete integrals [1, 2], the general sampling-by-counting reduction requires a large number of such estimates (proportional to the number of variables) for each sample. Further, the accuracy guarantees on the sampling probability quickly become loose when taking ratios of estimates. In contrast, PAWS is a more direct and practical sampling approach, providing better accuracy guarantees while requiring a much smaller number of NP-oracle queries per sample.

Answering NP-oracle queries, of course, requires exponential time in the worst case, in accordance with the hardness of sampling. We rely on the fact that combinatorial search tools, however, are often extremely fast in practice, and any complete solver can be used as a black box in our sampling scheme. Another key advantage is that when combinatorial search succeeds, our analysis provides a certificate that, with high probability, any samples produced will be distributed within an (arbitrarily small) constant factor of the desired probability distribution. In contrast, with MCMC methods it is generally hard to assess whether the chain has mixed. We empirically demonstrate that PAWS outperforms MCMC as well as variational methods on hard synthetic Ising Models and on a real-world test case generation problem for software verification.

## 2 Setup and Problem Definition

We are given a probability distribution $p$ over a (high-dimensional) discrete set $\mathcal{X}$, where the probability of each item $x \in \mathcal{X}$ is proportional to a weight function $w : \mathcal{X} \to \mathbb{R}^+$, with $\mathbb{R}^+$ being the set of non-negative real numbers. Specifically, given $x \in \mathcal{X}$, its probability $p(x)$ is given by

$$p(x) = \frac{w(x)}{Z} \ , \quad Z = \sum_{x \in \mathcal{X}} w(x)$$

where $Z$ is a normalization constant known as the *partition function*. We assume $w$ is specified compactly, e.g., as the product of factors or in a conjunctive normal form. As our driving example, we consider the case of undirected discrete graphical models [10] with $n = |V|$ random variables $\{x_i, i \in V\}$ where each $x_i$ takes values in a finite set $\mathcal{X}_i$. We consider a factor graph representation for a joint probability distribution over elements (or *configurations*) $x \in \mathcal{X} = \mathcal{X}_1 \times \cdots \times \mathcal{X}_n$:

$$p(x) = \frac{w(x)}{Z} = \frac{1}{Z} \prod_{\alpha \in \mathcal{I}} \psi_\alpha(\{x\}_\alpha). \tag{1}$$

This is a compact representation for $p(x)$ based on the weight function $w(x) = \prod_{\alpha \in \mathcal{I}} \psi_\alpha(\{x\}_\alpha)$, defined as the product of potentials or factors $\psi_\alpha : \{x\}_\alpha \mapsto \mathbb{R}^+$, where $\mathcal{I}$ is an index set and $\{x\}_\alpha \subseteq V$ the subset of variables factor $\psi_\alpha$ depends on. For simplicity of exposition, without loss of generality, we will focus on the case of binary variables, where $\mathcal{X} = \{0, 1\}^n$.

We consider the fundamental problem of (approximately) **sampling** from $p(x)$, i.e., designing a randomized algorithm that takes $w$ as input and outputs elements $x \in \mathcal{X}$ according to the probability distribution $p$. This is a hard computational problem in the worst case. In fact, it is more general than NP-complete decision problems (e.g., sampling solutions of a SATisfiability instance specified as a factor graph entails finding at least one solution, or deciding there is none). Further, samples can be used to approximate #P-complete problems [2], such as estimating a marginal probability.

## 3 Sampling by Embed, Project, and Search

Conceptually, our sampling strategy has three steps, described in Sections 3.1, 3.2, and 3.3, resp. (1) From the input distribution $p$ we construct a new distribution $p'$ that is "close" to $p$ but more

discrete. Specifically, $p'$ is based on a new weight function $w'$ that takes values only in a discrete set of geometrically increasing weights. (2) From $p'$, we define a *uniform* probability distribution $p''$ over a carefully constructed higher-dimensional **embedding** of $\mathcal{X} = \{0,1\}^n$. The previous discretization step allows us to specify $p''$ in a compact form, and sampling from $p''$ can be seen to be precisely equivalent to sampling from $p'$. (3) Finally, we indirectly sample from the desired distribution $p$ by **sampling uniformly** from $p''$, by randomly projecting the embedding onto a lower-dimensional subspace using universal hash functions and then searching for feasible states.

The first and third steps involve a bounded loss of accuracy, which we can trade off with computational efficiency by setting hyper-parameters of the algorithm. A key advantage is that *our technique reduces the weighted sampling problem to that of solving one MAP query (i.e., finding the most likely state) and a polynomial number of feasibility queries (i.e., finding any state with non-zero probability)* for the original graphical model augmented (through an embedding) with additional variables and carefully constructed factors. In practice, we use a combinatorial optimization package, which requires exponential time in the worst case (consistent with the hardness of sampling) but is often fast in practice. Our analysis shows that whenever the underlying combinatorial search and optimization queries succeed, the samples produced are **guaranteed**, with high probability, to be coming from an approximately accurate distribution.

### 3.1 Weight Discretization

We use a geometric discretization of the weights into "buckets", i.e., a uniform discretization of the log-probability. As we will see, $\Theta(n)$ buckets are sufficient to preserve accuracy.

**Definition 1.** Let $M = \max_x w(x)$, $r > 1$, $\epsilon > 0$, and $\ell = \lceil \log_r(2^n/\epsilon) \rceil$. Partition the configurations into the following weight based disjoint *buckets*: $\mathcal{B}_i = \{x \mid w(x) \in \left( \frac{M}{r^{i+1}}, \frac{M}{r^i} \right] \}, i = 0, \ldots, \ell-1$ and $\mathcal{B}_\ell = \{x \mid w(x) \in [0, \frac{M}{r^\ell}] \}$. The *discretized weight function* $w' : \{0,1\}^n \to \mathbb{R}^+$ is defined as follows: $w'(x) = \frac{M}{r^{i+1}}$ if $x \in \mathcal{B}_i$ for $i < \ell$ and $w'(x) = 0$ if $x \in \mathcal{B}_\ell$. The corresponding *discretized probability distribution* $p'(x) = w'(x)/Z'$ where $Z'$ is the normalization constant.

**Lemma 1.** *Let $\rho = r^2/(1-\epsilon)$. For all $x \in \cup_{i=0}^{l-1} \mathcal{B}_\ell$, $p(x)$ and $p'(x)$ are within a factor of $\rho$ of each other. Furthermore, $\sum_{x \in \mathcal{B}_\ell} p(x) \leq \epsilon$.*

*Proof.* Since $w$ maps to non-negative values, we have $Z \geq M$. Further,

$$\sum_{x \in \mathcal{B}_\ell} p(x) = \frac{1}{Z} \sum_{x \in \mathcal{B}_\ell} w(x) \leq \frac{1}{Z} |\mathcal{B}_\ell| \frac{M}{r^\ell} = \frac{|\mathcal{B}_\ell|}{2^n} \frac{\epsilon M}{Z} \leq \frac{\epsilon M}{Z} \leq \epsilon.$$

This proves the second part of the claim. For the first part, note that by construction, $Z' \leq Z$ and

$$Z' = \sum_{i=0}^{\ell} \sum_{x \in \mathcal{B}_i} w'(x) \geq \sum_{i=0}^{\ell-1} \sum_{x \in \mathcal{B}_i} \frac{1}{r} w(x) = \frac{1}{r} \left( Z - \sum_{x \in \mathcal{B}_\ell} w(x) \right) \geq (1-\epsilon)Z.$$

Thus $Z$ and $Z'$ are within a factor of $r/(1-\epsilon)$ of each other. For all $x$ such that $w(x) \notin \mathcal{B}_n$, recalling that $r > 1 > 1 - \epsilon$ and that $w(x)/r \leq w'(x) \leq rw(x)$, we have

$$\frac{1}{\rho} p(x) \leq \frac{w(x)}{rZ} \leq \frac{w(x)}{rZ'} \leq \frac{w'(x)}{Z'} = p'(x) = \frac{w'(x)}{Z'} \leq \frac{rw(x)}{Z'} \leq \frac{r^2}{1-\epsilon} \frac{w(x)}{Z} = \rho p(x).$$

This finishes the proof that $p(x)$ and $p'(x)$ are within a factor of $\rho$ of each other. $\square$

**Remark 1.** If the weights $w$ defined by the original graphical model are represented in finite precision (e.g., there are $2^{64}$ possible weights in double precision floating point), for every $b \geq 1$ there is a possibly large but *finite* value of $\ell$ (such that $M/r^\ell$ is smaller than the smallest representable weight) such that $\mathcal{B}_\ell$ is empty and the discretization error $\epsilon$ is effectively zero.

### 3.2 Embed: From Weighted to Uniform Sampling

We now show how to reduce the problem of sampling from the discrete distribution $p'$ (weighted sampling) to the problem of uniformly sampling, without loss of accuracy, from a higher-dimensional discrete set into which $\mathcal{X} = \{0,1\}^n$ is embedded. This is inspired by slice sampling [8], and can be intuitively understood as its *discrete* counterpart where we uniformly sample points $(x, y)$ from a discrete representation of the area under the ($y$ vs. $x$) probability density function of $p'$.

**Definition 2.** Let $w : \mathcal{X} \to \mathbb{R}^+$, $M = \max_x w(x)$, and $r = 2^b/(2^b - 1)$. Then the embedding $\mathcal{S}(w, \ell, b)$ of $\mathcal{X}$ in $\mathcal{X} \times \{0, 1\}^{(\ell-1)b}$ is defined as:

$$\mathcal{S}(w, \ell, b) = \left\{ (x, y_1^1, y_1^2, \ldots, y_{\ell-1}^{b-1}, y_{\ell-1}^b) \,\Big|\, w(x) \leq \frac{M}{r^i} \Rightarrow \bigvee_{k=1}^b y_i^k, 1 \leq i \leq \ell - 1; w(x) > \frac{M}{r^\ell} \right\}.$$

where $\bigvee_{k=1}^b y_i^k$ may alternatively be thought of as the linear constraint $\sum_{k=1}^b y_i^k \geq 1$. Further, let $p''$ denote a uniform probability distribution over $\mathcal{S}(w, \ell, b)$ and $n' = n + (\ell - 1)b$.

Given a compact representation of $w$ within a combinatorial search or optimization framework, the set $\mathcal{S}(w, \ell, b)$ can often be easily encoded using the disjunctive constraints on the $y$ variables.

**Lemma 2.** *Let* $(x, y) = (x, y_1^1, y_1^2, \cdots, y_1^b, y_2^1, \cdots, y_2^b, \cdots, y_{\ell-1}^1 \cdots, y_{\ell-1}^b)$ *be a sample from* $p''$, *i.e., a uniformly sampled element from* $\mathcal{S}(w, \ell, b)$. *Then* $x$ *is distributed according to* $p'$.

Informally, given $x \in \mathcal{B}_i$ and $x' \in \mathcal{B}_{i+1}$ with $i + 1 \leq l - 1$, there are precisely $r = 2^b/(2^b - 1)$ times more valid configurations $(x, y)$ than $(x', y')$. Thus $x$ is sampled $r$ times more often than $x'$. A formal proof may be found in the Appendix.

### 3.3  Project and Search: Uniform Sampling with Hash Functions and an NP-oracle

In principle, using the technique of Bellare et al. [3] and $n'$-wise independent hash functions we can sample purely *uniformly* from $\mathcal{S}(w, \ell, b)$ using an NP oracle to answer feasibility queries. However, such hash functions involve constructions that are difficult to implement and reason about in existing combinatorial search methods. Instead, we use a more practical algorithm based on pairwise independent hash functions that can be implemented using *parity constraints* (modular arithmetic) and still provides accuracy guarantees. The approach is similar to [5], but we include an algorithmic way to estimate the number of parity constraints to be used. We also use the pivot technique from [6] but extend that work in two ways: we introduce a parameter $\alpha$ (similar to [5]) that allows us to trade off uniformity against runtime and also provide *upper bounds* on the sampling probabilities.

We refer to our algorithm as PArity-basedWeightedSampler (PAWS) and provide its pseudocode as Algorithm 1. The idea is to *project* by randomly constraining the configuration space using a family of universal hash functions, *search* for up to $P$ "surviving" configurations, and then, if fewer than $P$ survive, perform rejection sampling to choose one of them. The number $k$ of constraints or factors (encoding a randomly chosen hash function) to add is determined first; this is where we depart from both Gomes et al. [5], who do not provide a way to compute $k$, and Chakraborty et al. [6], who do not fix $k$ or provide upper bounds. Then we repeatedly add $k$ such constraints, check whether fewer than $P$ configurations survive, and if so output one configuration chosen using rejection sampling. Intuitively, we need the hashed space to contain no more than $P$ solutions because that is a base case where we know how to produce uniform samples via enumeration. $k$ is a guess (accurate with high probability) of the number of constraints that is likely to reduce (by hashing) the original problem to a situation where enumeration is feasible. If too many or too few configurations survive, the algorithm fails and is run again. The small failure probability, accounting for a potentially poor choice of random hash functions, can be bounded irrespective of the underlying graphical model.

A combinatorial *optimization* procedure is used once in order to determine the maximum weight $M$ through MAP inference. $M$ is used in the discretization step. Subsequently, several *feasibility queries* are issued to the underlying combinatorial search procedure in order to, e.g., count the number of surviving configurations and produce one as a sample.

We briefly review the construction and properties of universal hash functions [11, 12].

**Definition 3.** $\mathcal{H} = \{h : \{0, 1\}^n \to \{0, 1\}^m\}$ is a *family of pairwise independent hash functions* if the following two conditions hold when a function $H$ is chosen uniformly at random from $\mathcal{H}$: 1) $\forall x \in \{0, 1\}^n$, the random variable $H(x)$ is uniformly distributed in $\{0, 1\}^m$; 2) $\forall x_1, x_2 \in \{0, 1\}^n$ $x_1 \neq x_2$, the random variables $H(x_1)$ and $H(x_2)$ are independent.

**Proposition 1.** *Let* $A \in \{0, 1\}^{m \times n}$, $c \in \{0, 1\}^m$. *The family* $\mathcal{H} = \{h_{A,c}(x) : \{0, 1\}^n \to \{0, 1\}^m\}$ *where* $h_{A,c}(x) = Ax + c \mod 2$ *is a family of pairwise independent hash functions.*

Further, $\mathcal{H}$ is also known to be a family of three-wise independent hash functions [5].

**Algorithm 1** Algorithm PAWS for sampling configurations $\sigma$ according to $w$

---

1: **procedure** COMPUTEK($n', \delta, P, \mathcal{S}$)
2:      $T \leftarrow 24 \lceil \ln (n'/\delta) \rceil$ ;    $k \leftarrow -1$ ;    $count \leftarrow 0$
3:      **repeat**
4:         $k \leftarrow k + 1$ ;    $count \leftarrow 0$
5:         **for** $t = 1, \cdots, T$ **do**
6:            Sample hash function $h_{A,c}^k : \{0,1\}^{n'} \rightarrow \{0,1\}^k$
7:            Let $\mathcal{S}^{k,t} \triangleq \{(x,y) \in \mathcal{S}, h_{A,c}^k(x,y) = 0\}$
8:            **if** $|\mathcal{S}^{k,t}| < P$ **then**                   `/* search for ` $\geq P$ ` different elements */`
9:               $count \leftarrow count + 1$
10:         **end for**
11:      **until** $count \geq \lceil T/2 \rceil$ or $k = n'$
12:      **return** $k$
13: **end procedure**

14: **procedure** PAWS($w : \{0,1\}^n \rightarrow \mathbb{R}^+, \ell, b, \delta, P, \alpha$)
15:      $M \leftarrow \max_x w(x)$          `/* compute with one MAP inference query on ` $w$ ` */`
16:      $\mathcal{S} \leftarrow \mathcal{S}(w, \ell, b)$ ;   $n' \leftarrow n + b(\ell - 1)$              `/* as in Definition 2 */`
17:      $i \leftarrow$ COMPUTEK($n', \delta, \gamma, P, \mathcal{S}$) $+ \alpha$
18:      Sample hash fn. $h_{A,c}^i : \{0,1\}^{n'} \rightarrow \{0,1\}^i$, i.e., uniformly choose $A \in \{0,1\}^{i \times n'}, c \in \{0,1\}^i$
19:      Let $\mathcal{S}^i \triangleq \{(x,y) \in \mathcal{S}, h_{A,c}^i(x,y) = 0\}$
20:      Check if $|\mathcal{S}^i| \geq P$ by searching for at least $P$ different elements
21:      **if** $|\mathcal{S}^i| \geq P$ or $|\mathcal{S}^i| = 0$ **then**
22:         **return** $\perp$                                              `/* failure */`
23:      **else**
24:         Fix an arbitrary ordering of $\mathcal{S}^i$              `/* for rejection sampling */`
25:         Uniformly sample $p$ from $\{0, 1, \ldots, P-1\}$
26:         **if** $p \leq |\mathcal{S}^i|$ **then**
27:            Select $p$-th element $(x,y)$ of $\mathcal{S}^i$ ;    **return** $x$
28:         **else**
29:            **return** $\perp$                                        `/* failure */`
30: **end procedure**

---

Lemma 3 (see Appendix for a proof) shows that the subroutine COMPUTEK in Algorithm 1 outputs with high probability a value close to $\log(|\mathcal{S}(w, \ell, b)|/P)$. The idea is similar to an unweighted version of the WISH algorithm [9] but with tighter guarantees and using more feasibility queries.

**Lemma 3.** *Let* $\mathcal{S} = \mathcal{S}(w, \ell, b) \subseteq \{0,1\}^{n'}$, $\delta > 0$, *and* $\gamma > 0$. *Further, let* $P \geq \min\{2, 2^{\gamma+2}/(2^{\gamma} - 1)^2\}$, $Z = |\mathcal{S}|$, $k_P^* = \log(Z/P)$, *and* $k$ *be the output of procedure* COMPUTEK($n', \delta, P, \mathcal{S}$). *Then,* $\mathbb{P}[k_P^* - \gamma \leq k \leq k_P^* + 1 + \gamma] \geq 1 - \delta$ *and* COMPUTEK *uses* $O(n' \ln (n'/\delta))$ *feasibility queries.*

**Lemma 4.** *Let* $\mathcal{S} = \mathcal{S}(w, \ell, b) \subseteq \{0,1\}^{n'}$, $\delta > 0$, $P \geq 2$, *and* $\gamma = \log\left((P + 2\sqrt{P+1} + 2)/P\right)$. *For any* $\alpha \in \mathbb{Z}$, $\alpha > \gamma$, *let* $c(\alpha, P) = 1 - 2^{\gamma - \alpha}/(1 - \frac{1}{P} - 2^{\gamma - \alpha})^2$. *Then with probability at least* $1 - \delta$ *the following holds:* PAWS($w, \ell, b, \delta, P, \alpha$) *outputs a sample with probability at least* $c(\alpha, P) 2^{-(\gamma+\alpha+1)} \frac{P}{P-1}$ *and, conditioned on outputting a sample, every element* $(x,y) \in \mathcal{S}(w, \ell, b)$ *is selected (Line 27) with probability* $p_s'(x,y)$ *within a constant factor* $c(\alpha, P)$ *of the uniform probability* $p''(x,y) = 1/|\mathcal{S}|$.

*Proof Sketch.* For lack of space, we defer details to the Appendix. Briefly, the probability $\mathbb{P}[\sigma \in \mathcal{S}^i]$ that $\sigma = (x,y)$ survives is $2^{-i}$ by the properties of the hash functions in Definition 3, and the probability of being selected by rejection sampling is $1/(P-1)$. Conditioned on $\sigma$ surviving, the mean and variance of the size of the surviving set $|\mathcal{S}^i|$ are independent of $\sigma$ because of 3-wise independence. When $k_P^* - \gamma \leq k \leq k_P^* + 1 + \gamma$ and $i = k + \alpha$, $\alpha > \gamma$, on average $|\mathcal{S}^i| < P$ and the size is concentrated around the mean. Using Chebychev's inequality, one can upper bound by $1 - c(\alpha, P)$ the probability $\mathbb{P}[S_i \geq P \mid \sigma \in \mathcal{S}^i]$ that the algorithm fails because $|S_i|$ is too large. Note that the bound is independent of $\sigma$ and lets us bound the probability $p_s(\sigma)$ that $\sigma$ is output:

$$c(\alpha, P) \frac{2^{-i}}{P-1} = \left(1 - \frac{2^{\gamma - \alpha}}{(1 - \frac{1}{P} - 2^{\gamma-\alpha})^2}\right) \frac{2^{-i}}{P-1} \leq p_s(\sigma) \leq \frac{2^{-i}}{P-1}. \qquad (2)$$

From $i = k + \alpha \leq k_P^* + 1 + \gamma + \alpha$ and summing the lower bound of $p_s(\sigma)$ over all $\sigma$, we obtain the desired lower bound on the success probability. Note that given $\sigma, \sigma'$, $p_s(\sigma)$ and $p_s(\sigma')$ are within a constant factor $c(\alpha, P)$ of each other from (2). Therefore, the probabilities $p_s'(\sigma)$ (for various $\sigma$) that $\sigma$ is output conditioned on outputting a sample are also within a constant factor of each other. From the normalization $\sum_\sigma p_s'(\sigma) = 1$, one gets the desired result that $p_s'(x, y)$ is within a constant factor $c(\alpha, P)$ of the uniform probability $p''(x, y) = 1/|\mathcal{S}|$. $\qquad \square$

### 3.4 Main Results: Sampling with Accuracy Guarantees

Combining pieces from the previous three sections, we have the following main result:

**Theorem 1.** *Let* $w : \{0, 1\}^n \to \mathbb{R}^+$, $\epsilon > 0$, $b \geq 1$, $\delta > 0$, *and* $P \geq 2$. *Fix* $\alpha \in \mathbb{Z}$ *as in Lemma 4,* $r = 2^b/(2^b - 1)$, $\ell = \lceil \log_r(2^n/\epsilon) \rceil$, $\rho = r^2/(1-\epsilon)$, *bucket* $\mathcal{B}_\ell$ *as in Definition 1, and* $\kappa = 1/c(\alpha, P)$. *Then* $\sum_{x \in \mathcal{B}_\ell} p(x) \leq \epsilon$ *and with probability at least* $(1 - \delta)c(\alpha, P)2^{-(\gamma+\alpha+1)}\frac{P}{P-1}$, PAWS($w$, $\ell$, $b$, $\delta$, $P$, $\alpha$) *succeeds and outputs a sample* $\sigma$ *from* $\{0, 1\}^n \setminus \mathcal{B}_\ell$. *Upon success, each* $\sigma \in \{0, 1\}^n \setminus \mathcal{B}_\ell$ *is output with probability* $p_s'(\sigma)$ *within a constant factor* $\rho\kappa$ *of the desired probability* $p(\sigma) \propto w(\sigma)$.

*Proof.* Success probability follows from Lemma 4. For $x \in \{0, 1\}^n \setminus \mathcal{B}_\ell$, combining Lemmas 1, 2, 4 we obtain

$$
\frac{1}{\rho\kappa}p(x) \leq \frac{1}{\kappa}p'(x) = \sum_{y:(x,y)\in\mathcal{S}(w,\ell,b)} \frac{1}{\kappa}p''(x,y) \leq \sum_{y|(x,y)\in\mathcal{S}(w,\ell,b)} p_s'(x,y) = p_s'(x)
$$

$$
\leq \sum_{y:(x,y)\in\mathcal{S}(w,\ell,b)} \kappa p''(x,y) = \kappa p'(x) \leq \rho\kappa p(x)
$$

where the first inequality accounts for discretization error from $p(x)$ to $p'(x)$ (Lemma 1), equality follows from Lemma 2, and the sampling error between $p''$ and $p_s'$ is bounded by Lemma 4. The rest is proved in Lemmas 1, 2. $\qquad \square$

**Remark 2.** By appropriately setting the hyper-parameters $b$ and $\ell$ we can make the discretization errors $\rho$ and $\epsilon$ arbitrarily small. Although this does not change the number of required feasibility queries, it can significantly increase the runtime of combinatorial search because of the increased search space size $|\mathcal{S}(w, \ell, b)|$. Practically, one should set these parameters as large as possible, while ensuring combinatorial searches can be completed within the available time budget. Increasing parameter $P$ improves the accuracy as well, but also increases the number of feasibility queries issued, which is proportional to $P$ (but does not affect the structure of the search space). Similarly, by increasing $\alpha$ we can make $\kappa$ arbitrarily small. However, the probability of success of the algorithm decreases exponentially as $\alpha$ is increased. We will demonstrate in the next section that a practical tradeoff between computational complexity and accuracy can be achieved for reasonably sized problems of interest.

**Corollary 2.** *Let* $w, b, \epsilon, \ell, \delta, P, \alpha$, *and* $\mathcal{B}_\ell$ *be as in Theorem 1, and* $p_s'(\sigma)$ *be the output distribution of* PAWS($w$, $\ell$, $b$, $\delta$, $P$, $\alpha$). *Let* $\phi : \{0, 1\}^n \to \mathbb{R}$ *and* $\eta_\phi = \max_{x \in \mathcal{B}_\ell} |\phi(x)| \leq \|\phi\|_\infty$. *Then,*

$$
\frac{1}{\rho\kappa}\mathbb{E}_{p_s'}[\phi] - \epsilon\eta_\phi \leq \mathbb{E}_p[\phi] \leq \rho\kappa\mathbb{E}_{p_s'}[\phi] + \epsilon\eta_\phi
$$

*where* $\mathbb{E}_{p_s'}[\phi]$ *can be approximated with a sample average using samples produced by PAWS.*

## 4 Experiments

We evaluate PAWS on synthetic Ising Models and on a real-world test case generation problem for software verification. All experiments used Intel Xeon 5670 3GHz machines with 48GB RAM.

### 4.1 Ising Grids Models

We first consider the *marginal computation* task for synthetic grid-structured Ising models with random interactions (attractive and mixed). Specifically, the corresponding graphical model has $n$ binary variables $x_i$, $i = 1, \cdots, n$, with single node potentials $\psi_i(x_i) = \exp(f_i x_i)$ and pairwise

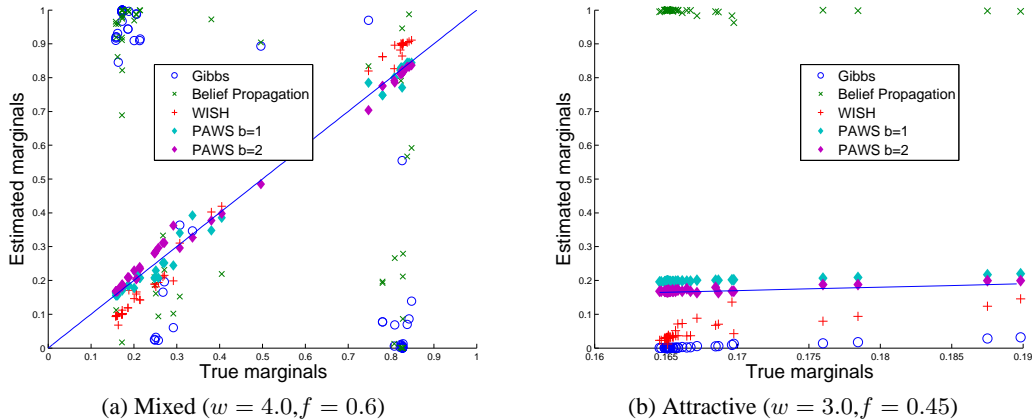

(a) Mixed ($w = 4.0, f = 0.6$)            (b) Attractive ($w = 3.0, f = 0.45$)

Figure 1: Estimated marginals vs. true marginals on $8 \times 8$ Ising Grid models. Closeness to the 45 degree line indicates accuracy. PAWS is run with $b \in \{1, 2\}$, $P = 4$, $\alpha = 1$, and $\ell = 25$ (mixed case) or $\ell = 40$ (attractive case).

interactions $\psi_{ij}(x_i, x_j) = \exp(w_{ij}x_ix_j)$, where $f_i \in_R [-f, f]$ and $w_{ij} \in_R [-w, w]$ in the *mixed* case, while $w_{ij} \in_R [0, w]$ in the *attractive* case.

Our implementation of PAWS uses the open source solver ToulBar2 [13] to compute $M = \max_x w(x)$ and as an oracle to check the existence of at least $P$ different solutions. We augmented ToulBar2 with the IBM ILOG CPLEX CP Optimizer 12.3 [14] based on techniques borrowed from [15] to efficiently reason about parity constraints (the hash functions) using Gauss-Jordan elimination. We run the subroutine COMPUTEK in Algorithm 1 only once at the beginning, and then generate all the samples with the same value of $i$ (Line 17). The comparison is with Gibbs sampling, Belief Propagation, and the recent WISH algorithm [9]. Ground truth is obtained using the Junction Tree method [16].

In Figure 1, we show a scatter plot of the estimated vs. true marginal probabilities for two Ising grids with mixed and attractive interactions, respectively, representative of the general behavior in the large-weights regime. Each sampling method is run for 10 minutes. Marginals computed with Gibbs sampling (run for about $10^8$ iterations) are clearly very inaccurate (far from the 45 degree line), an indication that the Markov Chain had not mixed as an effect of the relatively large weights that tend to create barriers between modes which are hard to traverse. In contrast, samples from PAWS provide much more accurate marginals, in part because it does not rely on local search and hence is not directly affected by the energy landscape (with respect to the Hamming metric). Further, we see that we can improve the accuracy by increasing the hyper-parameter $b$. These results highlight the practical value of having accuracy guarantees on the quality of the samples after finite amounts of time vs. MCMC-style guarantees that hold only after a potentially exponential mixing time.

Belief Propagation can be seen from Figure 1 to be quite inaccurate in this large-weights regime. Finally, we also compare to the recent WISH algorithm [9] which uses similar hash-based techniques to estimate the partition function of graphical models. Since producing samples with the general sampling-by-counting reduction [1, 2] or estimating each marginal as the ratio of two partition functions (with and without a variable clamped) would be too expensive (requiring $n + 1$ calls to WISH) we heuristically run it once and use the solutions of the optimization instances it solves in the inner loop as samples. We see in Figure 1 that while samples produced by WISH can sometimes produce fairly accurate marginal estimates, these estimates can also be far from the true value because of an inherent bias introduced by the $\arg\max$ operator.

## 4.2 Test Case Generation for Software Verification

Hardware and software verification tools are becoming increasingly important in industrial system design. For example, IBM estimates $100 million savings over the past 10 years from hardware verification tools alone [17]. Given that complete formal verification is often infeasible, the paradigm of choice has become that of randomly generating "interesting" test cases to stress the code or chip

| Instance | Vars | Factors | Time (s) | MSE ($\times 10^{-5}$) |
|---|---|---|---|---|
| bench1039 | 330 | 785 | 1710 | 5.76 |
| bench431 | 173 | 410 | 34.97 | 4.35 |
| bench115 | 189 | 458 | 52.75 | 20.74 |
| bench97 | 170 | 401 | 67.03 | 45.57 |
| bench590 | 244 | 527 | 593.71 | 8.11 |
| bench105 | 243 | 524 | 842.35 | 8.56 |

(a) Marginals: runtime and mean squared error

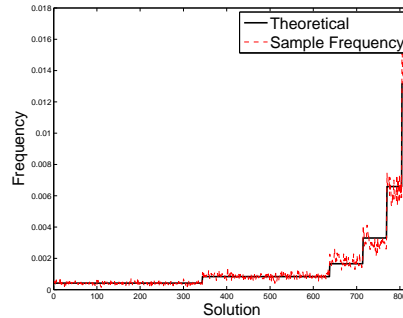

(b) True vs. observed sampling frequencies.

Figure 2: Experiments on software verification benchmark.

with the hope of uncovering bugs. Typically, a model based on *hard constraints* is used to specify consistent input/output pairs, or valid program execution traces. In addition, in some systems, domain knowledge can be specified by experts in the form of *soft constraints*, for instance to introduce a preference for test cases where operands are zero and bugs are more likely [17].

For our experiments, we focus on software (SW) verification, using an industrial benchmark [18] produced by Microsoft's SAGE system [19, 20]. Each instance defines a uniform probability distribution over certain valid traces of a computer program. We modify this benchmark by introducing soft constraints defining a weighted distribution over valid traces, indicating traces that meet certain criteria should be sampled more often. Specifically, following Naveh et al. [17] we introduce a preference towards traces where certain registers are zero. The weight is chosen to be a power of two, so that there is no loss of accuracy due to discretization using the previous construction with $b = 1$.

These instances are very difficult for MCMC methods because of the presence of very large regions of zero probability that cannot be traversed and thus can break the ergodicity assumption. Indeed we observed that Gibbs sampling often fails to find a non-zero probability state, and when it finds one it gets stuck there, because there might not be a non-zero probability path from one feasible state to another. In contrast, our sampling strategy is not affected and does not require any ergodicity assumption. Table 2a summarizes the results obtained using the propositional satisfiability (SAT) solver CryptoMiniSAT [21] as the feasibility query oracle for PAWS. CryptoMiniSAT has built-in support for parity constraints $Ax = c \mod 2$. We report the time to collect 1000 samples and the Mean Squared Error (MSE) of the marginals estimated using these samples. We report results only on the subset of instances where we could enumerate all feasible states using the exact model counter Relsat [22] in order to obtain ground truth marginals for MSE computation. We see that PAWS scales to fairly large instances with hundreds of variables and gives accurate estimates of the marginals. Figure 2b shows the theoretical vs. observed sampling frequencies (based on 50000 samples) for a small instance with 810 feasible states (execution traces), where we see that the output distribution $p'_s$ is indeed very close to the target distribution $p$.

## 5 Conclusions

We introduced a new approach, called PAWS, to the fundamental problem of sampling from a discrete probability distribution specified, up to a normalization constant, by a weight function, e.g., by a discrete graphical model. While traditional sampling methods are based on the MCMC paradigm and hence on some form of local search, PAWS can leverage more advanced combinatorial search and optimization tools as a black box. A significant advantage over MCMC methods is that PAWS comes with a strong accuracy guarantee: whenever combinatorial search succeeds, our analysis provides a certificate that, with high probability, the samples are produced from an approximately correct distribution. In contrast, accuracy guarantees for MCMC methods hold only in the limit, with unknown and potentially exponential mixing times. Further, the hyper-parameters of PAWS can be tuned to trade off runtime with accuracy. Our experiments demonstrate that PAWS outperforms competing sampling methods on challenging domains for MCMC.

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
