[Supplementary Material]

# A  Appendix: Proofs

*Proof of Lemma 2.* The discretized set of possible weights for $p'(x)$ is $w' \in \left\{0, \frac{M}{r^\ell}, \cdots, \frac{M}{r}\right\}$. Rescaling by a factor of $r2^{b(\ell-1)}/M$, this is equivalent to a rescaled set of weights

$$w'' \in \left\{0, (2^b - 1)^{\ell-1}, 2^b(2^b-1)^{\ell-2}, \cdots, 2^{(\ell-2)b}(2^b - 1), 2^{(\ell-1)b}\right\}$$

For any $i = 0 \cdots, \ell - 1$, let $x \in \mathcal{B}_i$. Then by definition there are precisely $(2^b - 1)^i 2^{(\ell-1-i)b}$ valid assignments of $(y_1^1, y_1^2, \cdots, y_1^b, y_2^1, \cdots, y_2^b, \cdots, y_{\ell-1}^1 \cdots, y_{\ell-1}^b)$ such that $(x,y) \in \mathcal{S}(w, \ell, b)$. Thus, $x$ is sampled with probability proportional to $w''(x)$ as desired. Now suppose $x \in \mathcal{B}_\ell$. Then $w(x) \leq \frac{M}{r^\ell}$, so $x$ is sampled with probability zero by definition of $\mathcal{S}(w, \ell, b)$ simply because there is no valid assignment to the $y$ variables such that $(x, y) \in \mathcal{S}(w, \ell, b)$. □

*Proof of Lemma 3.* Let $T \leftarrow 24 \lceil \ln(n'/\delta) \rceil$ as in Algorithm 1. For $t \in \{1, \ldots, T\}$, let $S_i^t = |\{(x, y) \in \mathcal{S} : h_{A,c}^i(x, y) = 0\}|$ be the number of elements of $\mathcal{S}$ that satisfy $h_{A,c}^i(x,y) = 0$, i.e., "survive" after adding $i$ random parity constraints. The output of COMPUTEK is nothing but

$$k = \min\left\{\min\left\{i \mid \text{Median}(S_i^1, \cdots, S_i^T) < P\right\}, \ n'\right\}$$

where the default value $n'$ is taken if the inner "min" is over an empty set. It follows from pairwise independence of the chosen hash functions that:

$$\mu_i \triangleq \mathbb{E}[S_i^t] = \frac{Z}{2^i}, \quad \sigma_i^2 \triangleq \text{Var}[S_i^t] = \frac{Z}{2^i}\left(1 - \frac{1}{2^i}\right)$$

For $i \leq k_P^*$, Chebychev inequality yields:

$$\mathbb{P}[S_i^t < P] \leq \mathbb{P}[|S_i^t - \mu_i| > (\mu_i - P)] \leq \frac{\sigma_i^2}{(\mu_i - P)^2} \leq \frac{Z/2^i}{(Z/2^i - P)^2}$$

The RHS is an increasing function of $i$, so for $i \leq k_P^* - \gamma$, which implies $Z/2^i \leq P2^\gamma$, we have $\mathbb{P}[S_i^t < P] \leq 2^\gamma/((2^\gamma - 1)^2 P) \triangleq 1 - q$. For $P \geq 2^{\gamma+2}/(2^\gamma - 1)^2$, we thus have $\mathbb{P}[S_i^t < P] \leq 1/4$ and $q \geq 3/4$. In other words, more than half the $S_i^t$ are expected to be at least as large as $P$. Using Chernoff inequality,

$$\mathbb{P}\left[\text{Median}(S_i^1, \cdots, S_i^T) \geq P\right] = 1 - \mathbb{P}\left[|\{t \mid S_i^t < P\}| < T/2\right] \geq 1 - \exp\left(-\frac{1}{2q}T\left(q - \frac{1}{2}\right)^2\right).$$

Similarly, for $i \geq k_P^* + \gamma$, we have $\mu_i < P$ and from Chebychev Inequality

$$\mathbb{P}[S_i^t \geq P] \leq \mathbb{P}[|S_i^t - \mu_i| \geq (P - \mu_i)] \leq \frac{\sigma_i^2}{(\mu_i - P)^2} \leq \frac{Z/2^i}{(Z/2^i - P)^2} \leq 2^\gamma/((2^\gamma - 1)^2 P) \leq \frac{1}{4}.$$

Using Chernoff inequality for $i \geq k_P^* + \gamma$,

$$\mathbb{P}\left[\text{Median}(S_i^1, \cdots, S_i^T) < P\right] \geq 1 - \exp\left(-\frac{1}{2q}T\left(q - \frac{1}{2}\right)^2\right).$$

Combining these two observations, we get that

$$\mathbb{P}\left[k_P^* - \gamma \leq \min\left\{i \mid \text{Median}(S_i^1, \cdots, S_i^T) < P\right\} \leq \lceil k_P^* + \gamma \rceil\right] \geq$$

$$\mathbb{P}\left[\bigcap_{i=1}^{\lfloor k_P^* - \gamma \rfloor} \left(\text{Median}(S_i^1, \cdots, S_i^T) \geq P\right) \bigcap \left(\text{Median}(S_{\lceil k_P^* + \gamma \rceil}^1, \cdots, S_{\lceil k_P^* + \gamma \rceil}^T) < P\right)\right] \geq$$

$$1 - n'\exp\left(-\frac{4}{6}T\left(\frac{3}{4} - \frac{1}{2}\right)^2\right) = 1 - n'\exp(-\beta T) \geq 1 - \delta$$

for $T \geq \frac{1}{\beta}\ln(n'/\delta)$ where $\beta = \frac{1}{24}$. It holds trivially that

$$k_P^* = \log Z - \log P \leq n' - \log P$$

so from $\lceil k_P^* + \gamma \rceil \leq 1 + k_P^* + \gamma$ we also get

$$\mathbb{P}[k_P^* - \gamma \leq k \leq 1 + k_P^* + \gamma] \geq 1 - \delta$$

This finishes the proof. □

*Proof of Lemma 4.* It can be verified that $\gamma = \log\left((P + 2\sqrt{P+1} + 2)/P\right)$ is the unique positive solution to $P = 2^{\gamma+2}/(2^\gamma - 1)^2$. Therefore, $\gamma$ and $P$ satisfy the conditions of Lemma 3. Let $k$ be the output of procedure COMPUTEK$(n', \delta, P, \mathcal{S})$. Then from Lemma 3, we have that $\mathbb{P}[k_P^* - \gamma \leq k \leq k_P^* + 1 + \gamma] \geq 1 - \delta$. All probabilities below are implicitly conditioned on this event. Let

$$S_i = |\{(x,y) \in \mathcal{S}(w, \ell, b), h_{A,c}^i(x,y) = 0\}| = |\mathcal{S}(w, \ell, b)^i| = |\mathcal{S}^i|$$

be the number of solutions suriving after adding $i$ random parity constraints. It follows from pairwise independence of the hash functions (Definition 3) that

$$\mu_i \triangleq \mathbb{E}[S_i] = \frac{Z}{2^i} \ , \ \sigma_i^2 \triangleq \mathrm{Var}[S_i] = \frac{Z}{2^i}\left(1 - \frac{1}{2^i}\right)$$

Let $\alpha \geq \gamma$ and $i = k + \alpha$. Then

$$\mu_{k+\alpha} = \frac{Z}{2^{k+\alpha}} \leq \frac{P}{2^{\alpha-\gamma}}$$

that is, on average we are left with less than $P$ elements after adding $i$ random parity constraints. Let $\sigma = (x, y) \in \mathcal{S}(w, \ell, b)$ be an element of the set we want to sample from. The probability $p_s(\sigma)$ that $\sigma$ is output is

$$p_s(\sigma) \triangleq \mathbb{P}\left[S_i < P, \sigma \in \mathcal{S}(w, \ell, b)^i\right]\frac{1}{P-1}$$

$$= \mathbb{P}\left[S_i < P \mid \sigma \in \mathcal{S}(w, \ell, b)^i\right]\mathbb{P}\left[\sigma \in \mathcal{S}(w, \ell, b)^i\right]\frac{1}{P-1}$$

where for any $\sigma$, $\mathbb{P}[\sigma \in \mathcal{S}(w, \ell, b)^i] = 2^{-i}$. Thus we have

$$p_s(\sigma) = \mathbb{P}[S_i < P \mid \sigma \in \mathcal{S}(w, \ell, b)^i]\frac{2^{-i}}{P-1} \tag{3}$$

Now the expected value of the size of the set (and its variance) conditioned on $\sigma \in \mathcal{S}(w, \ell, b)^i$ are independent of $\sigma$ because of three-wise independence [5]. So we have

$$\mathbb{E}[S_i \mid \sigma \in \mathcal{S}(w, \ell, b)^i] = 1 + \frac{(Z-1)}{2^i} = \mu_i(\sigma)$$

$$\mathrm{Var}[S_i \mid \sigma \in \mathcal{S}(w, \ell, b)^i] = \frac{(Z-1)}{2^i}\left(1 - \frac{1}{2^i}\right) < \mathbb{E}[S_i \mid \sigma \in \mathcal{S}(w, \ell, b)^i]$$

We first note that $(Z-1)/2^i < Z/2^i = Z/2^{k+\alpha} \leq Z/2^{k_P^*-\gamma+\alpha} = P2^{\gamma-\alpha}$. Using Chebychev's inequality

$$\mathbb{P}[S_i \geq P \mid \sigma \in \mathcal{S}(w, \ell, b)^i] \leq \mathbb{P}[|S_i - \mu_i(\sigma)| \geq (P - \mu_i(\sigma)) \mid \sigma \in \mathcal{S}(w, \ell, b)^i]$$

$$\leq \frac{\frac{(Z-1)}{2^i}\left(1 - \frac{1}{2^i}\right)}{(P - (1 + \frac{(Z-1)}{2^i}))^2} \leq \frac{P2^{\gamma-\alpha}\left(1 - \frac{1}{2^i}\right)}{(P - 1 - P2^{\gamma-\alpha})^2} \leq \frac{2^{\gamma-\alpha}}{(1 - \frac{1}{P} - 2^{\gamma-\alpha})^2} \triangleq 1 - c(\alpha, P)$$

Plugging into (3) we get

$$c(\alpha, P)\frac{2^{-i}}{P-1} = \left(1 - \frac{2^{\gamma-\alpha}}{(1 - \frac{1}{P} - 2^{\gamma-\alpha})^2}\right)\frac{2^{-i}}{P-1} \leq p_s(\sigma) \leq \frac{2^{-i}}{P-1} \tag{4}$$

where $c(\alpha, P) \to 1$ as $\alpha \to \infty$. This shows that the sampling probabilities $p_s(\sigma)$ and $p_s(\sigma')$ of $\sigma$ and $\sigma'$, respectively, must be within a constant factor $c(\alpha, P)$ of each other.

From $k \leq k_P^* + 1 + \gamma$ it follows that

$$p_s(\sigma) \geq c(\alpha, P)\frac{2^{-(1+\gamma+\alpha)}}{Z}\frac{P}{P-1}$$

This shows that the probability that the algorithm does not output $\perp$ is at least

$$\mathbb{P}[\text{output} \neq \perp] = Q = \sum_{\sigma \in \mathcal{S}(w, \ell, b)} p_s(\sigma) \geq c(\alpha, P)2^{-(1+\gamma+\alpha)}\frac{P}{P-1}$$

The probability $p'_s(\sigma)$ that $\sigma$ is sampled given that the algorithm does not output $\perp$ is

$$\frac{\mathbb{P}\left[S_i < P, \sigma \in \mathcal{S}(w, \ell, b)^i, \text{output} \neq \perp\right]}{Q} = \frac{\mathbb{P}\left[S_i < P, \sigma \in \mathcal{S}(w, \ell, b)^i\right]}{Q} = \frac{p_s(\sigma)}{Q} = p'_s(\sigma)$$

Plugging in (4)

$$c(\alpha, P)\frac{2^{-i}}{P-1}\frac{1}{Q} \leq p'_s(\sigma) \leq \frac{2^{-i}}{P-1}\frac{1}{Q}$$

From $\sum_\sigma p'_s(\sigma) = 1$ we get

$$c(\alpha, P)\frac{2^{-i}}{P-1}\frac{1}{Q}Z \leq 1 \leq \frac{2^{-i}}{P-1}\frac{1}{Q}Z$$

which implies

$$c(\alpha, P)\frac{1}{Z} \leq c(\alpha, P)\frac{2^{-i}}{P-1}\frac{1}{Q} \leq p'_s(\sigma) \leq \frac{2^{-i}}{P-1}\frac{1}{Q} \leq \frac{1}{c(\alpha, P)}\frac{1}{Z}$$

This finishes the proof. $\qquad\square$

*Proof of Corollary 2.* Suppose we want to compute an expectation of $\phi : \{0,1\}^n \to \mathbb{R}$

$$\mathbb{E}_p[\phi] = \sum_{x \in \{0,1\}^n} p(x)\phi(x) = \sum_{x \in \{0,1\}^n \backslash \mathcal{B}_\ell} p(x)\phi(x) + \sum_{x \in \mathcal{B}_\ell} p(x)\phi(x)$$

$$\sum_{x \in \{0,1\}^n \backslash \mathcal{B}_\ell} p(x)\phi(x) - \epsilon\eta_\phi \leq \mathbb{E}_p[\phi] \leq \sum_{x \in \{0,1\}^n \backslash \mathcal{B}_\ell} p(x)\phi(x) + \epsilon\eta_\phi$$

From Theorem 1

$$\sum_{x \in \{0,1\}^n \backslash \mathcal{B}_\ell} \frac{1}{\rho\kappa}p(x)\phi(x) \leq \mathbb{E}_{p'_s}[\phi] = \sum_{x \in \{0,1\}^n \backslash \mathcal{B}_\ell} p'_s(x)\phi(x) \leq \sum_{x \in \{0,1\}^n \backslash \mathcal{B}_\ell} \rho\kappa p(x)\phi(x)$$

It follows that

$$\frac{1}{\rho\kappa}\mathbb{E}_{p'_s}[\phi] - \epsilon\eta_\phi \leq \mathbb{E}_p[\phi] \leq \rho\kappa\mathbb{E}_{p'_s}[\phi] + \epsilon\eta_\phi$$

as desired. $\qquad\square$