[Reviews · NeurIPS 2013]

Submitted by Assigned_Reviewer_4

This paper introduces a novel sampling algorithm for graphical models. The algorithm creates an auxiliary distribution close to the desired distribution by discretizing the weights. A higher-dimensional embedding is defined using this distribution such that uniform samples from this embedding correspond to the samples from the auxiliary distribution. The uniform samples are generated using projection using hash function, followed by a search procedure. Experiments on synthetic grids and a software verification application demonstrate better marginals as compared to MCMC and BP.

The proposed approach is quite interesting and potentially very useful. Although the approach extends upon a number of existing approaches, its application for inference in discrete graphical model is quite novel. I also like the idea of using combinatorial solvers as black boxes to solve harder problems such as marginal inference. The experimental results were also quite impressive for the synthetic grids.

I have the following reservations with the paper:

- The writing in the paper was quite confusing and unclear. The authors rely too much on notation, instead of describing the approach. A running example would've been helpful.

- The paper does not concisely describe the related work, instead it is strewn across multiple sections.

- The authors describe how the various parameters of the algorithm (b,l) control the tradeoff between accuracy and speed. It would've been useful to examine this tradeoff as these parameters are varied empirically.

- The author do a poor job of describing the fail cases of the algorithm. For example, if the optimizer fails to find sufficient feasible points, the sampler does not propose a sample (rejection sampling style). But the authors do not discuss what would lead to a better or worse rejection rate, i.e. is it based on the distribution of weights, the size of the configuration space, the factorization of the model, etc.
Summary: The paper introduces a sampling approach for discrete graphical models that is accompanied by strong theoretical bounds and significantly outperforms MCMC and BP on synthetic grids. I feel this would be a valuable contribution for NIPS.

Submitted by Assigned_Reviewer_5

This paper presents a new approach to sampling from binary graphical models. There are two main tricks in this paper. The first is a clever construction to transform an arbitrary distribution over binary vectors x into a uniform distribution over an expanded space. This is done by augmenting x with additional bits, which are subject to parity constraints that ensure that the number of is approximately proportional to the original probability of x. This is a uniform distribution over a set with complex nonlinear constraints, so sampling from this distribution is challenging. Hence the second trick, to sample from this distribution by adding random parity constraints, running combinatorial search to find configurations of x that satisfy the random constraints, and returns the feasible points.
By controlling the parameters of this procedure carefully, the authors obtain guarantees that this procedure approximately samples from the given disribution.

The method requires access to two subroutines: One to perform MAP inference in the model, and one that can find feasible configurations of x that satisfy the random parity constraints. Both of these are nonlinear optimization problems, the complexity of which depends on the complexity of the model.

To my knowledge, this is an interesting and creative approach which could be useful for situations in which there are strong
correlations between the variables, which can cause problems for simpler methods like Gibbs sampling or belief propagation.

The experimental results are sufficient to indicate that this method is feasible for some practical problems, but they do leave unanswered questions about
in what situations the method is most effective. For example, in the synthetic results, the point of synthetic experiments, when they are well done, is to answer
questions about which inference methods are most appropriate for different families of distributions. The experiments in this paper don't provide any insight
into this. As far as I can tell, they show results from only two weight settings of the Ising model. (Can the authors confirm?) Certainly BP and Gibbs don't work this
poorly on all distributions, or else no one would use them. The results on the software verification benchmark are interesting and welcome, but it would be nice
to have more of a sense of what the state of the art is on this benchmark.

One practical disadvantage to this method is that it has five tuning parameters, and it could be difficult for users to understand interactions between them. Also, there is a (user-controllable) probability that the method will fail to output a sample; presumably, the user would simply re-run the procedure until it succeeds. Speculatively, I wonder if it's possible to "reparameterize" the tuning parameters into ones that are more interpretable? For example, suppose the user chooses only epsilon and rho*kappa and will re-run PAWS until it produces a sample. Could the other parameters be selected to minimize the number of total number of calls to the feasibility oracle?

Minor points:

* Line 156: This sentence seems slightly garbled.
* Line 352: Running for a fixed number of time is a fair comparison, but it would be useful to know how many iterations the Gibbs sampelr was run for.
* Line 361: Did belief propagation converge on these models?
Summary: To my knowledge, this is an interesting and creative approach which could be useful for situations in which there are strong
correlations between the variables, which can cause problems for simpler methods like Gibbs sampling or belief propagation.

Submitted by Assigned_Reviewer_6

The paper presents and evaluates a new method to sample from complex high-dimensional distributions by reducing the problem to a small number of related optimization and enumeration problems. It is shown that with high probability the sampling distribution is within a constant factor from the true distribution. Experimental results demonstrate cases where the proposed method outperforms standard MCMC and variational methods.

The work continues on a great series of earlier studies on how sampling and counting problems can be solved by using solvers for SAT and other hard combinatorial problems.
The approach is well motivated by the fact that existing solvers tend to be very fast on typical instances encountered in practice, even if the problems are NP-hard and the solvers' worst-case runtime guarantees look rather hopeless. Prior works in this research direction (after the early works of Valiant, Vazirani, and Jerrum) are chiefly by Gomes, Sabharwal, Selman et al.:


1. Carla P. Gomes, Ashish Sabharwal, Bart Selman:
Near-Uniform Sampling of Combinatorial Spaces Using XOR Constraints. NIPS 2006: 481-488

2. Carla P. Gomes, Ashish Sabharwal, Bart Selman:
Model Counting: A New Strategy for Obtaining Good Bounds. AAAI 2006: 54-61

3. Carla P. Gomes, Jörg Hoffmann, Ashish Sabharwal, Bart Selman:
Short XORs for Model Counting: From Theory to Practice. SAT 2007: 100-106

4. Carla P. Gomes, Jörg Hoffmann, Ashish Sabharwal, Bart Selman:
From Sampling to Model Counting. IJCAI 2007: 2293-2299

5. Carla P. Gomes, Willem Jan van Hoeve, Ashish Sabharwal, Bart Selman:
Counting CSP Solutions Using Generalized XOR Constraints. AAAI 2007: 204-209

6. Stefano Ermon, Carla P. Gomes, Bart Selman:
Uniform Solution Sampling Using a Constraint Solver As an Oracle. UAI 2012: 255-264

7. Stefano Ermon, Carla P. Gomes, Ashish Sabharwal, Bart Selman:
Taming the Curse of Dimensionality: Discrete Integration by Hashing and Optimization. ICML 2013

8. Stefano Ermon, Carla P. Gomes, Ashish Sabharwal, Bart Selman:
Optimization With Parity Constraints: From Binary Codes to Discrete Integration. UAI 2013


While the series as a whole constitutes a major innovation in the counting and sampling methodology, the present contribution is rather incremental. The main message of the paper, namely that combinatorial optimization and search tools enable sampling from complex distributions with controllable accuracy guarantees, follows already from the WISH algorithm given in the ICML'13 paper (7), since it is well known (e.g., due to Jerrum, Valiant, Vazirani) that accurate counting enables accurate sampling (supposing the problem is self-reducible in a way they often are). The proposed algorithm, PAWS, however does not implement that reduction: PAWS takes, in a sense, a more direct route to turn optimization and solution enumeration tools into a sampling algorithm. Only a single optimization instance needs to be solved, under the original combinatorial domain, which seems like a significant advantage. The downside is that one has to solve several enumeration instances in a much larger and more complicated domain. Under what conditions is PAWS more efficient than WISH (whether for sampling or integration)? The paper does not give even a preliminary answer to this question. In particular, the experiments reported in Figure 1 concern a variant of WISH that faster but potentially much less accurate than the original WISH.

The presentation is mostly excellent. The title and the abstract are particularly well written and informative, close to perfect. In Sections 3.2 and 3.3 there is, however, room for improvement: Def2 is difficult to follow at first glance. The description of the algorithm around Lemmas 3 and 4 gets dense and almost impossible to follow for a reader that is not familiar with the cited works [3,5,6]. While I appreciate the rigorous statements of Lemmas 3 and 4, I'd like to see an expanded description of the key ideas. Why is it so interesting whether the hashed space contains at least P solutions? What is the role of k? To get more space, I'd consider removing Corollary 2 and the Conclusions section.


- I'd use a bold font for emphasizing purposes only when defining new concepts.
- Just before Def1 I'd avoid "log-likelihood" because of the association to the likelihood function (of model parameters) that assumes some data are around.
- Sentences that end with a displayed equation should end with a ".".
- In Def2 I'd write write M/r^\ell < w(x) \leq M/r^i .
- In Section 3.3 use the reference numbers, e.g. [6], as parenthetical information, removing of which leaves an intelligent sentence.
- Consider placing the caption of Figure 2 below the figure.
Summary: A little high-quality advancement upon a fantastic series of works on a fundamental problem.
Author Feedback

Author rebuttal: Thanks all the reviewers for the constructive feedback!

Reviewer 1:

We’ll try to improve the presentation, consolidating the related work section and including a running example if space permits.

Parameter tuning is indeed an important aspect that we are currently exploring. See also discussion below on failure probability and parameter tuning.

We note that the small failure probability bounds hold for *all* graphical models. The failure probability in Theorem 1 accounts for a potential poor choice of random hash functions, not a choice of a graphical model. While staying within bounds, the failure probability could still vary a little depending on the interaction between the family of hash functions and the graphical model. We used a 3-wise independent family of hash functions, so in principle a pathological graphical model could be constructed adversarially so as to exploit 4-wise correlations between “heavy” configurations that jointly survive a randomly chosen hash function from this family. Nevertheless, the guaranteed bounds continue to hold irrespective of the graphical model.

Reviewer 2:

The two weights setting reported in the paper exemplify a general behavior in the large-weights case. That regime is notoriously hard for Gibbs and BP (the larger the weights, the more deterministic the model becomes). Indeed, Gibbs and BP perform much better in the easier low-weights regime (e.g., if the target distribution is almost uniform). In contrast, our method provides fairly accurate samples regardless of the weights, as long as combinatorial optimization and search are feasible.

Correct, in case of failure (which occurs with bounded probability) the algorithm is simply run again (rejection sampling style). We are also looking into re-parameterizations that would expose fewer parameters to the user. Devising effective ways to tune the parameters (perhaps adaptively) to improve performance (accuracy, number of calls but also their runtime, which is difficult to predict a priori and is also parameter-dependent) is a very interesting research direction that we are currently exploring.

We’ll clarify that Gibbs was run for about 10^8 iterations and BP did converge.

Reviewer 3:

Indeed PAWS can be seen as a more direct approach to sampling using combinatorial optimization and search. We will discuss more in detail the relationship with WISH, which was proposed as a method to compute (discrete) integrals.
It is true that in theory one can sample by taking ratios of partition functions. The main drawback of that approach is that one would need $Theta(n)$ calls to WISH (where n is the number of variables) to produce *a single* sample. In contrast, using PAWS after the initial estimation of k (which is roughly as expensive as one run of WISH), then each sample “only” costs one combinatorial search (albeit, in a more intricate space). Running WISH for 10 minutes to estimate each partition function as suggested in (7) was already impractical for sampling in our experiments (requiring hours to produce a single sample) . We therefore compared to a more practical WISH-based heuristic with a comparable runtime .
Another drawback is that even when WISH can solve all the optimizations to optimality and produces a guaranteed 16-approximation on Z, the accuracy guarantees on the *sampling probability* quickly become loose when taking ratios of estimates (e.g., factor 256 on each ratio).
When one has access to combinatorial search/optimization tools, we think PAWS is a more practical sampling approach than the general sampling-by-counting theoretical technique by Jerrum, Valiant, Vazirani.

Thanks for the suggestions on In Sections 3.2 and 3.3. We’ll improve the presentation emphasizing more the intuitive aspects.
Intuitively, we need the hashed space to contain no more than P solutions because that is a “base case” where we know how to produce uniform samples via enumeration. k is a guess (accurate with high probability) of the number of constraints needed to reduce (by hashing) the original problem to a situation where enumeration is feasible.